# Significance of Initial Serum Phosphate Imbalance in Traumatic Brain Injury with and Without Concomitant Spinal Injuries: Retrospective Analysis

**DOI:** 10.3390/biomedicines13122858

**Published:** 2025-11-24

**Authors:** Ayman El-Menyar, Ahammed Mekkodathil, Naushad A. Khan, Mohammad Asim, Bellal Joseph, Hassan Al-Thani

**Affiliations:** 1Clinical Research, Trauma and Vascular Surgery, Hamad Medical Corporation, Doha 3050, Qatar; amekkodathil@hamad.qa (A.M.); nkhan13@hamad.qa (N.A.K.); masim1@hamad.qa (M.A.); 2Department of Clinical Medicine, Weill Cornell Medical College, Doha 24144, Qatar; 3Department of Surgery, College of Medicine, The University of Arizona, Tucson, AZ 85721, USA; bjoseph@arizona.edu; 4Department of Surgery, Trauma and Vascular Surgery, Hamad Medical Corporation, Doha 3050, Qatar; halthani@hamad.qa

**Keywords:** serum phosphate, hypophosphatemia, hyperphosphatemia, traumatic brain injury, spinal injury, mortality, neurological deficit

## Abstract

**Objectives**: On-admission phosphate imbalance (OAPI) in traumatic brain injury (TBI) is scarce in the literature, either alone or with concomitant spinal injuries (CSI). We aimed to explore the OAPI in TBI and hypothesized that OAPI has unfavorable outcomes in TBI as well as spinal injury. **Methods**: In this retrospective study, all hospitalized patients with TBI were reviewed, and their serum phosphate levels were measured upon admission. Outcomes included in-hospital mortality and neurological deficit. **Results**: Among 912 TBI patients, 13% had hyperphosphatemia (*n* = 118/912) and 30% had associated CSI (*n* = 272/912). Hypophosphatemia was found in two-thirds of TBI patients (*n* = 607/912). Thirteen patients of CSI group developed neurological deficits (4.8%) with hypophosphatemia. Serum phosphate levels were significantly correlated with serum potassium, magnesium, and lactate levels, as well as Injury Severity Score (ISS). The serum glucose-phosphate ratio was higher in patients with severe Glasgow Coma Scale (GCS). The overall mortality was 21.3% (47% had hyperphosphatemia, 17% had hypophosphatemia, and 18% had normophosphatemia). Multivariable analysis showed that hyperphosphatemia, high serum lactate, sodium, and potassium levels, high head Abbreviated Injury Scale (AIS), and low GCS were significantly associated with increased mortality. **Conclusions**: Hypophosphatemia was common in TBI patients regardless of the presence of spinal injuries and was observed in all patients with neurological deficits. Routine phosphate monitoring may help in early risk stratification and targeted management of TBI.

## 1. Introduction

Traumatic brain injury (TBI) is a major cause of morbidity and mortality, often resulting in long-term disabilities. A subset of TBI patients also sustain concomitant spinal injuries, known as craniospinal injuries (CSIs) [1,2,3]. The prevalence of spinal injuries in TBI patients is estimated at 12%, with reported ranges between 1.6% and 11.4% [4,5,6,7,8,9,10]. CSIs are particularly severe, as they increase the risk of complications. Evidence suggests that older age groups are more prone to CSIs from vehicle crashes, while falling from height results in consistent CSI odds across all age groups [6]. Common traumatic brain lesions associated with CSIs include cerebral contusion, subarachnoid hematoma, and skull fractures [8].

TBI and CSI trigger complex physiological responses, including a heightened immune response, particularly in the spinal cord, compared to the brain [11]. Delayed diagnosis of CSIs can significantly worsen long-term patient outcomes [11]. Electrolyte imbalances are a well-documented complication in TBI patients, with studies highlighting their prevalence and impact on clinical outcomes [12,13,14,15,16,17]. Patients with spinal injuries, including those with CSIs, may have an elevated risk of electrolyte disturbances due to tissue and muscle destruction, prolonged immobility, muscle paralysis, renal impairment, and cardiovascular complications [18]. These imbalances can exacerbate the already critical condition of TBI and CSI patients.

Phosphorus, the second most abundant mineral in the body, is a critical intracellular anion vital for metabolic processes and energy production, including Adenosine Triphosphate (ATP) and 2,3-diphosphoglycerate (2,3-DPG) [19,20,21]. Polytrauma rapidly depletes ATP, underscoring the importance of phosphorus homeostasis [20]. Despite its significance, serum phosphate is less routinely measured or interpreted than sodium and potassium in trauma patients [15]. Hypophosphatemia in trauma patients can worsen respiratory muscle weakness, leading to difficulty during mechanical ventilation and weaning [15]. Severe hypophosphatemia may impair diaphragmatic contractility and cardiac function and contribute to neurologic dysfunction [19]. However, the causes and risk factors for abnormal phosphate levels following trauma, particularly in TBI and CSI patients, remain poorly understood.

The relationship between serum phosphate levels and injury severity is a critical area of focus due to its potential to guide patient management. While electrolyte imbalances in TBI have been studied, research on phosphate imbalances remains limited. This study aimed to evaluate the prevalence and significance of on-admission phosphate imbalance (OAPI) in patients with TBI, including those with spinal injuries (CSI), and to explore its potential role as a biomarker for predicting mortality. We hypothesized that OAPI negatively impacted the hospital course and outcome after TBI with and without spinal injury.

## 2. Materials and Methods

A retrospective study was conducted on TBI patients admitted to the Hamad Trauma Center (HTC) between 2016 and 2021. The HTC is the only level 1 trauma center in the state of Qatar providing free treatment for TBI injuries. Data were obtained from the Qatar National Trauma Registry (QTR) and the electronic medical records (CERNER). The QTR is regularly validated internally and externally and is linked to the American College of Surgeons-Trauma Quality Improvement Program (ACS-TQIP) and the National Trauma Data Bank (NTDB) [22]. Ethical approval with a waiver of consent for the study was obtained from the Institutional Review Board (MRC#01-21-501). 

We included all consecutive patients diagnosed with TBI if they had radiological (CT scan) spinal injuries (i.e., cervical, thoracic, or lumbar) and had a serum phosphate level measured on admission (the reading within the first 24 h). We excluded patients with penetrating injuries, those transferred from other hospitals, and any cases lacking phosphate value at the intensive care unit (ICU) admission.

Patients were admitted to HTC and treated according to the Advanced Trauma Life Support (ATLS) guidelines. Patients were initially categorized into CSI and non-CSI groups. CSI comprises concomitant brain and spine injury and is defined according to the ICD-10-CM codes, whereas the non-CSI group was patients with only TBI. Blood samples were collected to measure serum electrolyte levels on admission, and normal serum phosphate (Ph) level was defined as 1.1 to 1.45 mmol/L. Based on the Ph level, the cohort was divided into normophosphatemia, hypophosphatemia, and hyperphosphatemia. The serum glucose–phosphate ratio was also calculated.

Demographic information, mechanism of injuries, type of head trauma, injury scores, serum electrolyte levels (phosphate, sodium, potassium, calcium, and magnesium), serum lactate levels, interventions, complications, and in-hospital outcomes were collected for analysis. 

The severity of TBI was assessed using the Glasgow Coma Scale (GCS) and Abbreviated Injury Scale (AIS). The GCS is defined as severe (3–8), moderate (9–12), and mild (13–15) for mild head injuries [23]. The AIS rates the severity of injuries of different body parts on a scale of 1 to 6, with minor (AIS = 1), moderate (AIS = 2), serious (AIS = 3), severe (AIS = 4), critical (AIS = 5), and non-survivable injuries (AIS = 6). The Injury Severity Score (ISS) provides an overall score of anatomical injuries by squaring the AIS of the three most seriously injured body regions and adding them [23]. The ISS varies from 0 to 75, with 1–8 representing minor injuries, 9–15 moderate, 16–24 serious, 25–74 critical, and 75 as non-survivable [23]. Spinal injuries, including cervical (CS), thoracic (TS), and lumbar spines (LS), and any neurological deficits were defined according to ICD-10. This study was conducted according to the Strengthening the Reporting of Observational Studies in Epidemiology (STROBE) Statement-checklist.

### Statistical Analysis 

Mean and standard deviation (SD) were calculated for continuous variables with a normal distribution, while median and interquartile range (IQR) were used for variables with skewed distributions. Comparative analyses between CSI and non-CSI were performed for demographic and clinical characteristics. Categorical variables were compared using Chi-square tests. Continuous variables were compared using Student t-tests, and non-parametric tests were performed whenever applicable. Statistical significance was considered for a 2-tailed p-value less than 0.05. Pearson correlation coefficient analysis was performed to investigate the association of initial serum phosphate levels with the other relevant parameters. Multivariable logistic regression analysis (Enter method) was performed to determine the odds ratio (OR) and confidence interval (95% CI) for mortality for clinically relevant laboratory variables (serum Ph, Mg, Ca, Na, K, and lactate), in addition to age, gender, ISS, AIS, and GCS. All statistical analyses were performed using the Statistical Package for the Social Sciences (SPSS), (Version 26.0; IBM Corp., Armonk, NY, USA).

## 3. Results

### 3.1. Population Characteristics

Among 912 patients who were admitted with TBI, 13% of them had hyperphosphatemia, two-thirds had hypophosphatemia, and 272 patients (30%) had concomitant spinal injuries (CSI group) (Figure 1, Figure 2 and Figure 3). 

The mean age was 32 years, and the vast majority were male (94%), with no significant difference in age or gender distribution between the CSI and non-CSI groups (Table 1). CSI patients had significantly higher injury severity (ISS: 30.5 vs. 25.4, *p* < 0.001) and lower Glasgow Coma Scores (GCS: 4.9 vs. 6.0, *p* = 0.001). Thirteen patients developed neurological deficits (4.8% of CSI). Extradural hemorrhage was more common in non-CSI cases (*p* = 0.02), while other TBI lesions were similarly distributed.

### 3.2. Electrolyte Imbalance

Hyperphosphatemia was less common (13% in TBI and 14% in CSI). Hypophosphatemia was evident in all patients with neurologic deficits. Compared to the non-CSI group, CSI patients had significantly higher sodium levels (142 vs. 141 mmol/L, *p* = 0.01) and lower calcium levels (1.9 vs. 2.0, *p* = 0.001). Phosphate levels were comparable *(p* = 0.11). Table 2 compares the phosphate levels among the TBI cohort. Patients with hyperphosphatemia were significantly younger, had greater ISS, and had higher mortality compared to the other phosphate groups.

### 3.3. TBI Severity by GCS 

Table 3 compares the severity of TBI based on the GCS. Patients with severe TBI were significantly younger (median 29 years), had greater ISS, fewer phosphate levels (0.92 (IQR 0.72–1.21 mmol/L)), a higher glucose-phosphate ratio (11.17 (SD 7.7), greater ventilatory days, prolonged ICU stay, and higher mortality (26.7% vs. 6.8% and 5.6%) compared to mild and moderate TBI.

### 3.4. Electrolyte–Phosphate Correlations

Table 4 shows that serum phosphate correlated with serum potassium (*r* = 0.27, *p* = 0.001), serum magnesium (0.38, *p* = 0.001), serum lactate (*r* = 0.30, *p* = 0.001), and ISS (*r* = 0.20, *p* = 0.001). Phosphate correlated with hospital length of stay in patients with CSI (*r* = 0.15, *p* = 0.01). 

### 3.5. Neurological Deficits and Spinal Injury Levels

In patients with neurological deficits, cervical spine injuries were most common (*n* = 10), including five isolated CS cases, followed by thoracic (*n* = 5) and lumbar involvement (*n* = 3). Most of these patients (*n* = 11) survived. Serum phosphate levels were significantly lower in CSI patients with neurological deficits (*n* = 13) than those without [0.84 (SD 0.17) vs. 1.07 (SD 0.56) mmol/L, *p* = 0.001]. Figure 4 shows the distribution of spinal injuries in TBI patients.

### 3.6. Mortality and Phosphate Association

Overall, 21.3% of patients died during hospitalization. Mortality was higher in CSI patients (24.6% vs. 19.8%, *p* = 0.106) (Table 1). As shown in Figure 2, mortality among CSI patients with hyperphosphatemia was notably high at 59%. A similar trend was observed in non-CSI patients, with 41% mortality in those with hyperphosphatemia.

### 3.7. Multivariable Regression Analysis

High serum phosphate (OR 1.67; 95% CI: 1.05–2.66; *p* = 0.02), high serum sodium (OR 1.09; 95% CI: 1.04–1.15; *p* = 0.001), high head AIS (OR 2.65; 95% CI: 1.91–3.69; *p* = 0.001), high serum lactate (OR 1.30; 95% CI: 1.19–1.41; *p* = 0.001), and lower GCS (OR 0.89; 95% CI: 0.83–0.96; *p* = 0.004) were predictors of mortality (Figure 5).

## 4. Discussion

The association between initial serum phosphate and the neurologic deficit and mortality in patients with TBI with or without spinal injury is not well-explored in the literature. In this study, 912 consecutive hospitalized patients with TBI were analyzed, and nearly one-third had CSIs, highlighting the clinical relevance of combined head and spine trauma. Two-thirds of the cohort had hypophosphatemia on admission. Most patients had high ISS (Median 27), which may bias the source of phosphate imbalance in TBI patients. In the present study, patients with CSI exhibited more severe injury profiles compared to non-CSI patients. Notably, electrolyte disturbances, particularly phosphate levels, emerged as notable markers of poor prognosis. Hyperphosphatemia was associated with higher mortality in both CSI and non-CSI groups, and lower phosphate levels were significantly linked to the presence of neurological deficits. These findings underscore the potential utility of serum phosphate as a prognostic biomarker in TBI patients, particularly those with concomitant spinal injuries. Kim et al. showed that in trauma patients, hyperphosphatemia, ISS of >15, and older age were associated with a higher 30-day mortality [21]. Moreover, Im et al., in 2024, showed an increase in serum phosphate level 24 h after the initial measurement that was associated with higher mortality in patients with severe trauma [24]. However, these two studies did not acknowledge the TBI patients in their findings. Meanwhile, Polderman et al. concluded that patients with severe TBI are at high risk of hypophosphatemia in addition to hypomagnesemia and hypokalemia, which could be due to the increase in the urinary loss of various electrolytes [25]. Also, in TBI patients, Qiu et al. showed that admission serum glucose–phosphate ratio was significantly associated with trauma severity (GCS score) and outcome (GOS score) [26]. Our study aligns with this finding as this ratio was significantly higher in severe GCS than in mild and moderate GCS groups. 

Although research on TBI and CSIs is expanding, phosphate levels remain inadequately studied in TBI and CSI patients. A systematic review by Ngatuvai et al. [27], evaluated 14 studies on electrolyte abnormalities in TBI patients, reporting that hypernatremia, hypokalemia, and hypocalcemia were linked to increased mortality. However, this review did not address serum phosphate imbalances.

Hyponatremia is prevalent in the acute phase of spinal cord trauma, affecting approximately 85% of cases [18]. Chavasiri et al. [28], reported hyponatremia in 44% of cervical spine injuries. Our study found higher sodium levels in CSI patients than non-CSI patients, suggesting a different electrolyte pattern possibly influenced by injury severity or fluid management. Also, hypocalcemia was the most common electrolyte imbalance across the TBI cohort and CSI subgroup.

Besides hyponatremia and hypocalcemia, phosphate imbalance can impact the central nervous system, leading to symptoms such as confusion, depression, or seizures [29]. Hypophosphatemia may impair diaphragmatic function, complicating ventilation and ventilator weaning. A recent study reported that hyperphosphatemia was associated with increased 30-day mortality in overall trauma patients, with an adjusted odds ratio of 2.67 for severe cases [21]. In our study, the mortality in CSI patients with hyperphosphatemia reached 59%, and Kim et al. supported this finding [21]. They stated that elevated phosphate levels reflect severe systemic injury and poor prognosis. Hyperphosphatemia was also linked to fewer ICU-free days and higher blood transfusion volumes [21]. Elevated phosphate levels during acute ischemic conditions, such as hemorrhagic shock, are associated with poor outcomes [30,31,32]. Kim et al. [21], noted that hyperphosphatemia correlated with elevated lactate levels, indicative of ischemic tissue injury.

Additionally, TBI patients may experience hypophosphatemia compared to non-TBI trauma patients, suggesting ongoing phosphate loss [18]. Rugg et al. [33], observed hyperphosphatasemia in polytrauma patients upon TICU admission, likely due to tissue hypoxia and injury causing phosphate leakage from damaged cells, which may lead to endothelial dysfunction and oxidative stress [21]. Collectively, these findings suggest that hyperphosphatemia may serve as a surrogate marker of injury burden and systemic stress rather than being a direct cause of mortality, particularly in patients with TBI and CSI.

Factors such as polyuria, cerebral salt loss, mannitol diuresis, and inappropriate ADH secretion may contribute to hypophosphatemia in TBI patients [25]. Phosphate retention has been noted in spinal cord injury, potentially due to hypoparathyroidism [34]. Magnesium deficiency may further exacerbate phosphate depletion, with lower magnesium levels observed in TBI patients than in those with multiple fractures [25]. This study suggests a potential association between low calcium levels and phosphate imbalances in TBI and CSI patients. However, low calcium was also present in patients with normal phosphate levels.

Despite numerous studies on TBI and CSIs [2,3,5,6,9,34,35,36,37,38], few have specifically addressed electrolyte imbalances, particularly phosphate, in these populations. These studies have provided valuable insights into the mechanisms, presentations, and outcomes of TBI and CSIs, but electrolyte disturbances remain a critical yet understudied area. This is among the first studies to systematically evaluate serum phosphate in the context of combining TBI and spinal injury, highlighting its prognostic potential, especially in relation to neurological deficits and mortality.

### Limitations

The retrospective, single-center design of this study introduces potential selection bias and result generalizability. Most cases were polytraumatic TBI with high ISS; therefore, studies on isolated TBI are needed to better understand the association between TBI and electrolyte imbalance. The timing and frequency of blood sampling after admission were not detailed. Prospective studies are required to determine optimal timing and follow-up after corrective interventions. Nutritional status and feeding formulas were not documented and out of the study scope. The nature or description of the spinal injuries was not captured in this study and was included under “radiologic spinal injury"; however, the spinal AIS was given. The inclusion criteria were not limited to spinal cord injury; however, we believe that studies on “spinal cord” injury with TBI will be more informative. Lastly, the differential diagnosis of phosphate imbalances should be considered during the hospital course for proper management. Measuring all electrolytes in trauma patients on arrival is a routine in our institution. Further sources of bias could be gender and age as most patients were male and young. However, this reflects the nature of trauma in the country [39].

## 5. Conclusions

Phosphate imbalance is of prognostic value in TBI patients. OAPI is common in TBI patients, including those with CSIs, following blunt trauma. Hyperphosphatemia is associated with increased mortality, while hypophosphatemia is more prevalent in CSI patients with neurological deficits. These findings underscore the importance of vigilant monitoring and management of phosphate levels in this trauma population. Further research is required to develop targeted interventions and enhance patient care for TBI and CSI patients.

## Figures and Tables

**Figure 1 biomedicines-13-02858-f001:**
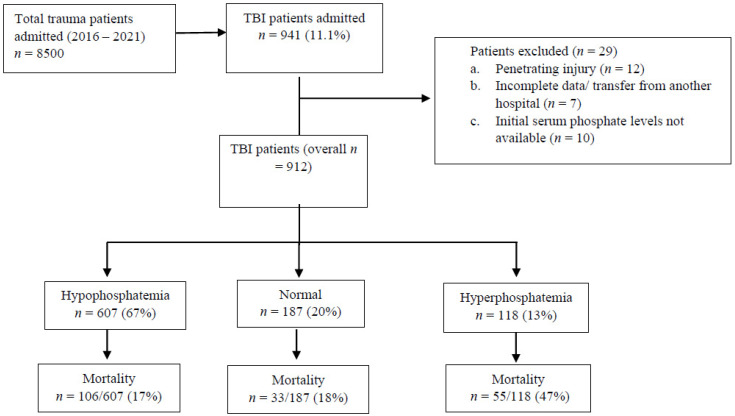
On-admission serum phosphate levels and mortality in traumatic brain injury (TBI) patients with or without spinal trauma.

**Figure 2 biomedicines-13-02858-f002:**
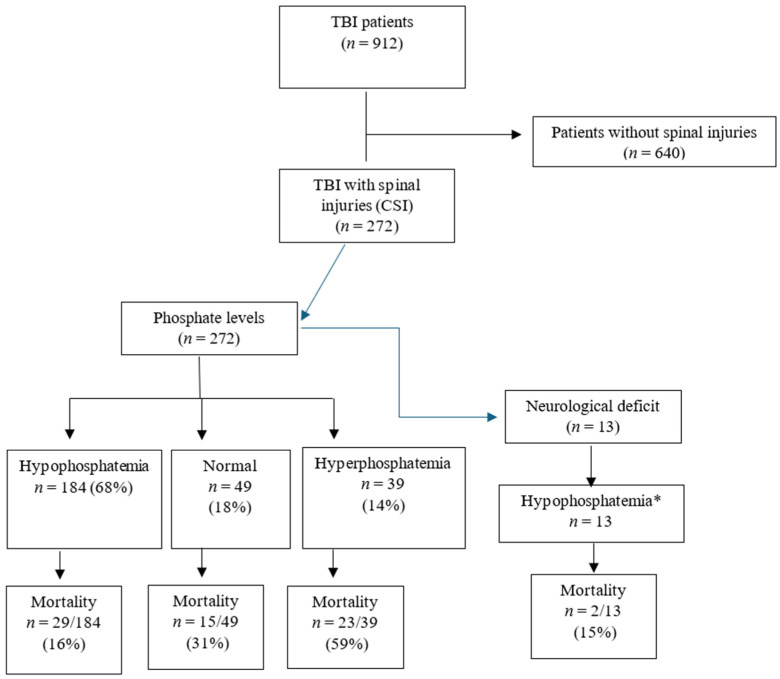
On-admission serum phosphate levels, mortality, and neurological deficit in traumatic brain injury (TBI) with spinal trauma (CSI). * No patients had normal or hyperphosphatemia.

**Figure 3 biomedicines-13-02858-f003:**
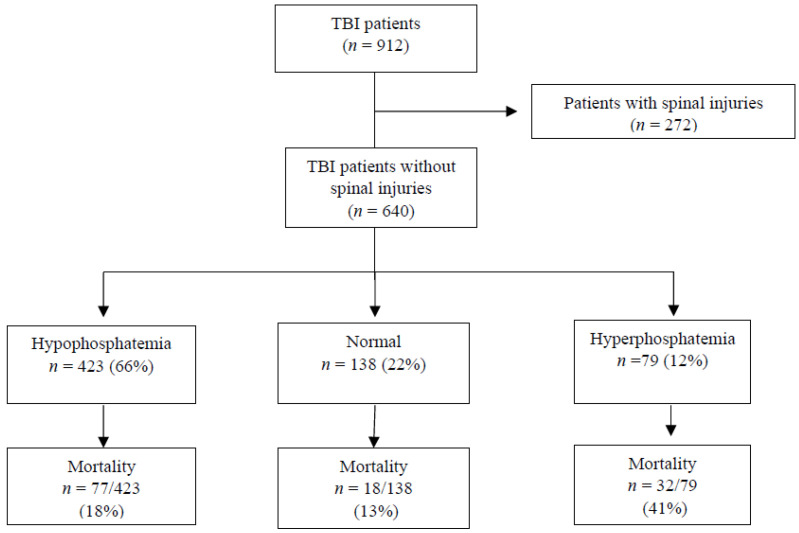
On-admission serum phosphate levels and mortality in traumatic brain injury (TBI) patients without spinal trauma.

**Figure 4 biomedicines-13-02858-f004:**
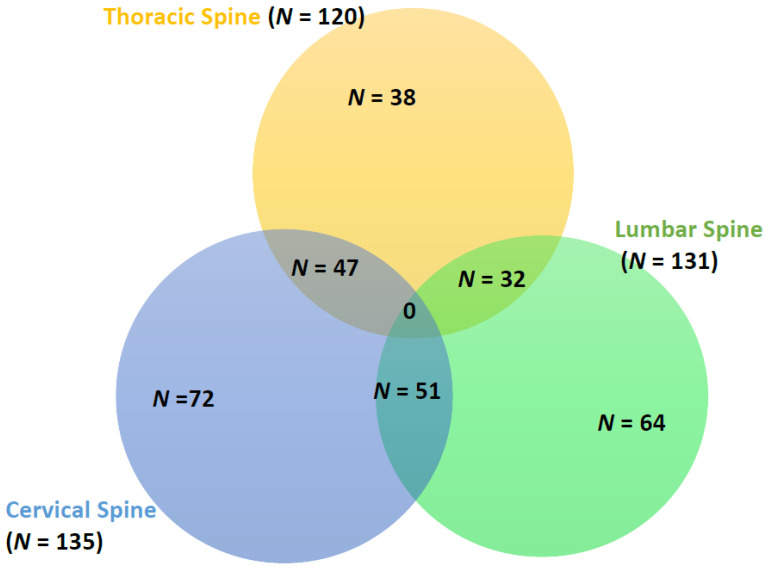
Distribution of spinal injuries.

**Figure 5 biomedicines-13-02858-f005:**
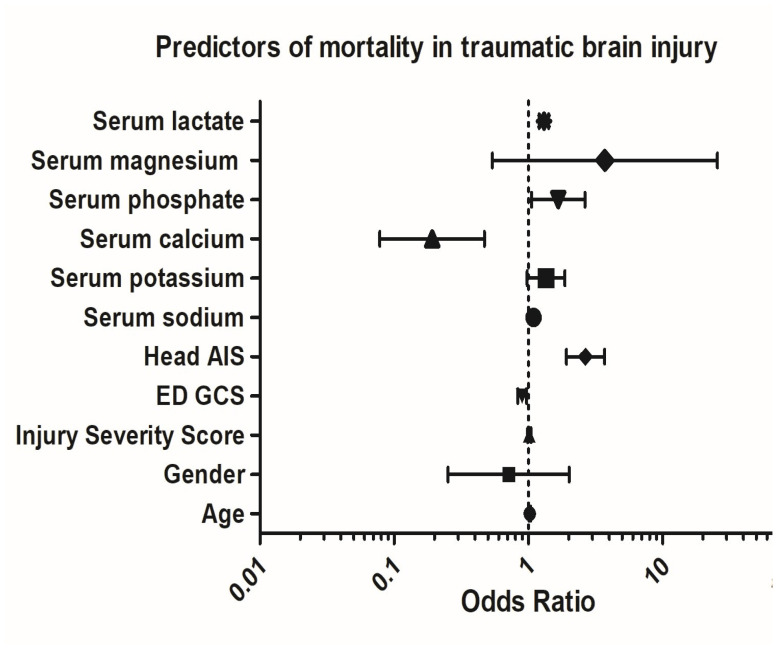
Predictors of mortality. (AIS: abbreviated injury scale; ED: emergency department; GCS: Glasgow coma scale).

**Table 1 biomedicines-13-02858-t001:** Demographic and clinical characteristics of traumatic brain injury (TBI) patients with (CSI) or without spinal injury (non-CSI).

Variable	Overall TBI (*n* = 912)	Non-CSI (*n* = 640)	CSI (*n* = 272)	*p* Value
Age (Median, IQR)	30 (2–40)	30 (22–41)	31 (23–39)	0.41
Males	853 (93.5%)	598 (93.4%)	255 (93.8%)	0.86
TBI type				
Subdural hemorrhage	319 (35.0%)	222 (34.7%)	97 (35.7%)	0.78
Subarachnoid hemorrhage	384 (42.1%)	262 (40.9%)	122 (44.9%)	0.27
Extradural hemorrhage	204 (22.4%)	157 (24.5%)	47 (17.3%)	0.02
Midline shift	206 (22.6%)	154 (24.1%)	52 (19.1%)	0.10
Neurological deficit	13 (1.4%)	-	13 (4.8%)	-
Glasgow Coma Score (Median, IQR)	3 (3–9)	3 (3–9)	3 (3–3)	0.001
Head AIS (Median, IQR)	4 (3–5)	4 (3–5)	4 (3–5)	0.73
Spine AIS (Median, IQR)	2 (2–2)	-	2 (2–2)	-
Injury Severity Score (Median, IQR)	27 (17–34)	26 (17–30)	29 (22–38)	0.001
Electrolyte levels (SD)				
Sodium (135–145 mmol/L)	141.0 (4.5)	140.7 (4.4)	141.7 (4.5)	0.01
Potassium (3.6–5.2 mmol/L)	3.8 (0.7)	3.8 (0.6)	3.9 (0.7)	0.28
Calcium (2.2–2.7 mmol/L)	1.9 (0.3)	2.0 (0.3)	1.9 (0.3)	0.001
Magnesium (0.65–1.05 mmol/L)	0.7 (0.1)	0.7 (0.1)	0.6 (0.1)	0.10
Phosphate (1.1–1.45 mmol/L)	1.0 (0.5)	1.0 (0.4)	1.1 (0.5)	0.11
Hemoglobin level (Median, IQR)	12.9 (11.2–14.3)	13.2 (11.4–14.4)	12.5 (10.8–13.9)	0.006
Mortality	194 (21.3%)	127 (19.8%)	67 (24.6%)	0.106

**Table 2 biomedicines-13-02858-t002:** Comparison of demographics, injury characteristics, and clinical outcomes by phosphate level categories in traumatic brain injury patients.

	Normophosphatemia (*n* = 187; 20.5%)	Hypophosphatemia (*n* = 607; 66.6%)	Hyperphosphatemia (*n* = 118; 12.9%)	*p* Value
Age *	28 (21–39)	31 (24–41)	26 (15–38)	0.04
Male	170 (90.9%)	579 (95.4%)	104 (88.1%)	0.004
Subdural hemorrhage	60 (32.1%)	224 (36.9%)	35 (29.7%)	0.21
Subarachnoid hemorrhage	88 (47.1%)	247 (40.7%)	49 (41.5%)	0.30
Extradural hemorrhage	45 (24.1%)	142 (23.4%)	17 (14.4%)	0.08
Midline shift	36 (19.3%)	139 (22.9%)	31 (26.3%)	0.34
Neurological deficit	-	13 (2.1%)	-	-
Glasgow Coma Score *	3 (3–11)	3 (3–8)	3 (3–6)	0.09
Cervical spine injury	26 (13.9%)	87 (14.3%)	22 (18.6%)	0.45
Thoracic spine injury	22 (11.8%)	83 (13.7%)	15 (12.7%)	0.79
Lumbar spine injury	24 (12.8%)	89 (14.7%)	18 (15.3%)	0.78
Injury Severity Score *	26 (17–34)	26 (17–33)	30 (26–38)	0.001
Ventilatory days *	4 (1–9)	4 (1–11)	3 (1–11)	0.59
ICU LOS *	8 (4–14)	8 (3–16)	5 (2–17)	0.07
Mortality	33 (17.6%)	106 (17.5%)	55 (46.6%)	0.001

* Median and interquartile range (IQR), ICU LOS: intensive care unit length of stay.

**Table 3 biomedicines-13-02858-t003:** Patient characteristics and outcomes by severity of traumatic brain injury (*n* = 903) *.

	Mild TBI (GCS 13–15) (*n* = 88; 9.7%)	Moderate TBI (GCS 9–12) (*n* = 144; 15.8%)	Severe TBI (GCS 3–8) (*n* = 671; 73.6%)	*p* Value
Age	33 (26–43)	31 (23–42)	29 (22–39)	0.006
Injury Severity Score	22 (16–29)	19 (14–26)	29 (22–35)	0.001
Serum phosphate level	1.03 (0.79–1.22)	0.97 (0.76–1.21)	0.92 (0.72–1.21)	0.001
Hypophosphatemia	60%	64%	68%	0.02
Hyperphosphatemia	8%	12%	14%	0.02
Serum glucose-phosphate ratio (SD)	9.34 (51)	9.34 (5.1)	11.17 (7.7)	0.004
Intubation	45 (51.1%)	97 (67.4%)	666 (99.3%)	0.001
Ventilatory days	1 (0–5)	1 (0–5)	6 (2–12)	0.001
TICU days	4 (2–11)	5 (3–11)	10 (4–17)	0.001
Mortality	6 (6.8%)	8 (5.6%)	179 (26.7%)	0.001

903 patients had GCS data *. Median and interquartile range (IQR) used for continuous variables.

**Table 4 biomedicines-13-02858-t004:** Pearson correlation coefficient analysis for initial serum phosphate level in traumatic brain injury.

Variables	Correlation and *p*-Value	Overall TBI	Non-CSI	CSI (*n* = 272)
Serum Sodium	Pearson Correlation	0.016	−0.008	0.048
	Sig. (2-tailed)	0.628	0.849	0.428
	*N*	912	640	272
Serum Potassium	Pearson Correlation	0.267	0.279	0.242
	Sig. (2-tailed)	0.001	0.001	0.001
	*N*	912	640	272
Serum Calcium	Pearson Correlation	0.000	0.020	−0.016
	Sig. (2-tailed)	0.997	0.620	0.795
	*N*	912	640	272
Serum Magnesium	Pearson Correlation	0.373	0.290	0.544
	Sig. (2-tailed)	0.001	0.001	0.001
	*N*	912	640	272
Cervical Spine AIS	Pearson Correlation	−0.103	-	−0.219
	Sig. (2-tailed)	0.234	-	0.090
	*N*	134	-	61
Thoracic Spine AIS	Pearson Correlation	−0.118	-	−0.155
	Sig. (2-tailed)	0.194	-	0.263
	*N*	122	-	54
Lumbar Spine AIS	Pearson Correlation	−0.097	-	−0.150
	Sig. (2-tailed)	0.273	-	0.293
	*N*	130	-	51
Injury severity score (ISS)	Pearson Correlation	0.202	0.159	0.270
	Sig. (2-tailed)	0.001	0.001	0.001
	*N*	912	640	272
Initial serum lactate level	Pearson Correlation	0.302	0.246	0.420
	Sig. (2-tailed)	0.001	0.001	0.001
	*N*	836	640	257

AIS: abbreviated injury scale, TBI: traumatic brain injury, CSI: craniospinal injury, *N* = number of patients, Sig: Significance.

## Data Availability

The original contributions presented in this study are included in the article. Further inquiries can be directed to the corresponding author(s).

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
