# Peer review of "Significance of Initial Serum Phosphate Imbalance in Traumatic Brain Injury with and Without Concomitant Spinal Injuries: Retrospective Analysis"

_biomedicines, 2025, doi:10.3390/biomedicines13122858_

Round 1

Reviewer 1 Report

Comments and Suggestions for Authors

   This study provides innovative insights into the significance of phosphate balance in patients with traumatic brain injury as well as in patients with spinal injury.  

Comments:

  1. Please check the font used for the numbers indicating the affiliations.
  2. For clarity of reading, it is suggested to introduce the full definition for ISS, AIS, and GCS.
  3. The percentage results in the Results section of the Abstract are difficult to follow, especially those in lines 24-27.
  4. The introduction should start on a separate page from the abstract.
  5. The authors introduced the abbreviation for the subarachnoid hematoma (SAH), but this abbreviation is not used elsewhere in the manuscript.
  6. Please define serum phosphate as Ph, not as pH level (line 96).
  7. Although this is a retrospective study with anonymized data for transparency, it is recommended to include the ethical approval number.
  8. Is it Abbreviated Injury Scale (AIS) (line 104) or Abbreviated injury score (AIS) (line 106)?
  9. Line 110: it would be clearer to simply use ISS, without adding a score.
  10. In the Figure 1, the letter "n" is italic consistently except under the C in the third inset.
  11. The font size in the legends of Figure 1 and Figure 2 is not consistent.
  12. Figure 2, make sure hypophosphatemia and hyperphosphatemia are on one line
  13. Figure 2 legend, extra space
  14. Please make the legends under Figure 1, Figure 2, and Figure 3 consistent, as the legend under Figure 2 includes the abbreviation for spinal injuries (CSI), while the others don't.
  15. In the Table 1 there is inconsistent spacing between the numbers and the values in the parentheses
  16. In the Table 1, "P value" is written with hyphen (P-value), while in the Table 2 and Table 3, "P value" is written without hyphen (P value)
  17. Line 172, please separate “SD7.7”.
  18. Lines 254 and 255: The sentence appears grammatically incorrect. The fragment starting “that elevated phosphate reflects…” should be connected properly to the previous sentence.
  19. Line 293, extra space.
  20. Besides hyponatremia and hypocalcemia (Line 249), both hypo- and hyperphosphatemia have been associated with mental illness and depression
  21. In the Discussion section (Lines 277-278), the authors state that this study contributes to understanding the mechanisms underlying TBI and CSI, which is preferable. However, the text provides limited insight into these mechanisms, as the results are presented more descriptively rather than mechanistically.
  22. The reference numbered 40 in the reference list is not properly formatted.

Author Response

  1. Please check the font used for the numbers indicating the affiliations. Done
  2. For clarity of reading, it is suggested to introduce the full definition for ISS, AIS, and GCS. Done
  3. The percentage results in the Results section of the Abstract are difficult to follow, especially those in lines 24-27. Number added for clarity
  4. The introduction should start on a separate page from the abstract. Done
  5. The authors introduced the abbreviation for the subarachnoid hematoma (SAH), but this abbreviation is not used elsewhere in the manuscript. Removed
  6. Please define serum phosphate as Ph, not as pH level (line 96). Corrected
  7. Although this is a retrospective study with anonymized data for transparency, it is recommended to include the ethical approval number. Added
  8. Is it Abbreviated Injury Scale (AIS) (line 104) or Abbreviated injury score (AIS) (line 106)?.  It is scale
  9. Line 110: it would be clearer to simply use ISS, without adding a score. Corrected
  10. In the Figure 1, the letter "n" is italic consistently except under the C in the third inset. Corrected
  11. The font size in the legends of Figure 1 and Figure 2 is not consistent. Corrected
  12. Figure 2, make sure hypophosphatemia and hyperphosphatemia are on one line: Revised
  13. Figure 2 legend, extra space: Done
  14. Please make the legends under Figure 1, Figure 2, and Figure 3 consistent, as the legend under Figure 2 includes the abbreviation for spinal injuries (CSI), while the others don't. Reply: abbreviation was given if it was mentioned in the figure boxes
  15. In the Table 1 there is inconsistent spacing between the numbers and the values in the parentheses: Spaces corrected
  16. In the Table 1, "P value" is written with hyphen (P-value), while in the Table 2 and Table 3, "P value" is written without hyphen (P value): hyphen removed
  17. Line 172, please separate “SD7.7”. corrected
  18. Lines 254 and 255: The sentence appears grammatically incorrect. The fragment starting “that elevated phosphate reflects…” should be connected properly to the previous sentence. Corrected
  19. Line 293, extra space. Space removed
  20. Besides hyponatremia and hypocalcemia (Line 249), both hypo- and hyperphosphatemia have been associated with mental illness and depression: added with a new ref
  21. In the Discussion section (Lines 277-278), the authors state that this study contributes to understanding the mechanisms underlying TBI and CSI, which is preferable. However, the text provides limited insight into these mechanisms, as the results are presented more descriptively rather than mechanistically. Reply: Actually, we were referring to studies in the literature , not ours: These studies have provided valuable insights into the mechanisms, presentations, and outcomes of TBI and CSIs, but electrolyte disturbances remain a critical yet understudied area. This is among the first studies to systematically evaluate serum phosphate in the context of combining TBI and spinal injury, highlighting its prognostic potential, especially in relation to neurological deficits and mortality.
  22. The reference numbered 40 in the reference list is not properly formatted. Revised

Reviewer 2 Report

Comments and Suggestions for Authors

The paper is well written, although the english could be improved.  Line 103: you can delete the description of GCS or replace it with a text box. It is also mandatory to insert the relative reference. 

The table 4 is too long and you can format it better. You can also insert a list of abbreviations. 

Finally, it is mandatory to give a conclusive message: for you is this electrolyte more prognostic or diagnostic?

Comments on the Quality of English Language

It could be improved in some expressions. All the scores must be cited with references. 

Author Response

The paper is well written, although the english could be improved.  Line 103: you can delete the description of GCS or replace it with a text box. It is also mandatory to insert the relative reference. : Thank you , done and references added

The table 4 is too long and you can format it better. You can also insert a list of abbreviations. : Table improved

Finally, it is mandatory to give a conclusive message: for you is this electrolyte more prognostic or diagnostic?: Reply, thank you, we added the prognostic value in the conclusion

Comments on the Quality of English Language: It could be improved in some expressions. All the scores must be cited with references. Done, edited

Round 2

Reviewer 1 Report

Comments and Suggestions for Authors

The authors of the manuscript have adequately addressed the requested corrections and suggestions following revision.